# Development of Palatable Amorphous Trazodone Hydrochloride Formulations via Ion Exchange

**DOI:** 10.3390/pharmaceutics17080972

**Published:** 2025-07-27

**Authors:** Zhaohua Li, Junjie Wang, Huijian Wang, Yibo Li, Qiang Fu

**Affiliations:** 1Wuya College of Innovation, Shenyang Pharmaceutical University, No. 103, Wenhua Road, Shenyang 110016, China; leezh0125spu@163.com (Z.L.); wjj13936379096@163.com (J.W.); 17859139969@163.com (H.W.); 18524454895@163.com (Y.L.); 2Joint International Research Laboratory of Intelligent Drug Delivery Systems, Ministry of Education, Shenyang 110016, China; 3NMPA Key Laboratory for Research and Evaluation of Pharmaceutical Preparations and Excipients, China Pharmaceutical University, Nanjing 210009, China

**Keywords:** ion exchange, amorphous, taste masking, bitterness, Amberlite IRP88

## Abstract

**Objectives:** The oral route is the most widely used method of administration. However, the bitter taste of drugs is a prevalent issue compromising patient acceptance. This study aimed to develop a palatable amorphous trazodone hydrochloride (TRA) formulation via ion exchange with Amberlite IRP88 resin as the carrier. **Methods:** TRA-Amberlite IRP88 complexes (TRCs) were prepared using the static exchange method and their physical properties were then characterized. Molecular docking was carried out to elucidate the molecular interaction. Finally, the dissolution profiles and taste of TRCs were evaluated. **Results:** The Physical characterizations confirmed that TRA was amorphously dispersed in Amberlite IRP88. Importantly, the in vivo taste masking study suggested that the bitterness of TRA was effectively masked. The reason was that the dissociation of TRCs was suppressed in the saliva, resulting in reduced dissolution in the oral cavity. **Conclusion:** this study suggests that amorphization is effective in masking the bitterness of drugs and provides guidance for the development of palatable oral formulations.

## 1. Introduction

The oral route is a preferred option for drug administration owing to its cost-effectiveness [1], non-invasiveness [2], and ease of administration [3]. However, patient acceptance is frequently diminished by the bitter taste of orally administered drugs [4,5,6]. Therefore, the development of taste-masked oral formulations is important to improve patient adherence and thereby ensure therapeutic efficacy.

In the pharmaceutical field, various approaches have been employed to mask the bitterness of drugs, including chemical modification [7], microencapsulation [8], and coating [9]. However, these methods are often constrained by complexity, high cost, or low efficiency. Recently, studies have reported that amorphization can mask the bitterness of drugs [10,11,12]. For example, Zhang et al. dispersed clarithromycin in Eudragit E100 using hot melt extrusion, demonstrating successful bitterness masking upon amorphization [10]. In another study, based on co-axial electrospinning technology, Abdelhakim et al. also effectively masked the unpleasant taste of chlorpheniramine maleate by preparing amorphous nanofibers using Eudragit^®^ EPO and Kollicoat^®^ Smartseal as carriers [11]. Furthermore, amorphization offers advantages such as simple preparation, cost-effectiveness, and solvent-free processing. Consequently, dispersing drugs within polymeric carriers to achieve an amorphous state represents an effective taste-masking strategy.

Ion-exchange resins (IERs) are water-insoluble polymeric matrices functionalized with acidic or basic groups [13,14,15]. Under certain conditions, these functional groups can reversibly exchange counterions with similarly charged, ionized drug molecules, forming amorphous drug–resin complexes [16,17,18]. These amorphous complexes exhibit resistance to dissociation in saliva [17,19], thereby preventing interaction between bitter drugs and taste receptors on the tongue [16,20,21,22]. Additionally, the ion-exchange process also has the advantages of simple preparation, cost-effectiveness, and solvent-free nature [20,21]. Therefore, the development of amorphous drug–resin complexes seems to be an effective strategy for taste-masking.

Amberlite IRP88 is a weakly acidic cation-exchange resin comprising a methacrylic acid–divinylbenzene copolymer matrix functionalized with potassium carboxylate groups (-COO^−^K^+^) as counterion binding sites [23]. During ion exchange, potassium ions (K^+^) on the Amberlite IRP88 resins can be reversibly exchanged with cationic drug molecules, forming amorphous drug–resin complexes (Figure 1) [24]. Moreover, these complexes can resist dissociation in saliva, thereby effectively reducing the dissolution of the drug in the oral cavity [20,24,25]. Therefore, Amberlite IRP88 is a promising carrier for preparing palatable amorphous formulations of cationic drugs [26,27].

Trazodone hydrochloride (TRA), a cationic drug and inhibitor of 5-HT2 receptors, is used for the treatment of depressive disorder [28]. In addition, it exhibits alpha-adrenergic-antagonist and moderate antihistamine effects, contributing to its hypnotic and anxiolytic actions [29]. Currently, the only approved dosage form for TRA is tablet. However, its bitter taste poses a significant challenge to patient compliance. Numerous studies have also reported that TRA could elicit taste or smell complaints [30,31,32]. Therefore, development of an effective taste-masking formulation is essential to improve the acceptability of TRA.

In this study, Amberlite IRP88 resin was employed as the carrier to develop a palatable amorphous formulation of TRA via ion exchange. TRA–resin complexes (TRCs) were prepared using the static exchange method. The physical properties of the TRCs were characterized with scanning electron microscopy tandem energy dispersive spectroscopy analysis (SEM-EDS), differential scanning calorimetry (DSC), powder X-ray diffraction (PXRD), and Fourier transform infrared spectroscopy (FT-IR). Molecular docking was performed to elucidate potential molecular interactions between TRA and Amberlite IRP88. Additionally, the dissolution profiles and taste-masking efficacy of TRCs were evaluated. These findings underscore the potential of amorphization via ion-exchange resins as an efficient taste-masking strategy for bitter drugs.

## 2. Materials and Methods

### 2.1. Materials

TRA was purchased from Xi’an Kono Chem Co., Ltd. (Xi’an, China). Amberlite IRP88 was provided as a gift by Dupont Company (Wilmington, DE, USA). Hydrochloric acid, acetic acid, sodium hydroxide, and potassium dihydrogen phosphate were purchased from Tianjin Hengxing Chemical Reagent Co., Ltd. (Tianjin, China).

### 2.2. Preparation of Drug–Resin Complexes

TRA–Amberlite IRP88 complexes (TRCs) with three different ratios (TRA: resin = 1:2, 1:1, and 2:1, *w*/*w*) were prepared using the batch method [33,34,35], which were coded as TRCs_1:2_, TRCs_1:1_, and TRCs_2:1_, respectively. Initially, TRA was dissolved in purified water to obtain 1.0 mg/mL drug solutions. Subsequently, the appropriate amount of Amberlite IRP88 was weighed and then slowly added to the TRA solutions. Next, the suspensions were maintained at 50 °C and stirred at 500 rpm using a WS-H280-Pro LED magnetic stirrer (DLAB Scientific Co., Ltd., Beijing, China) for 2 h. The complexes were collected by vacuum filtration, washed with deionized water to remove any unbound drugs and ions, and finally dried until they reached a constant weight. The concentration of TRA was determined at 246 nm using a T-6 UV spectrophotometer (Nanjing Philes Instrument Co., Ltd., Nanjing, China).

The drug loading (*DL*%), complexation efficiency (C*E*%), degree of drug loading (*F*%), and yield (*Y*%) were calculated using Equations (1)–(4), respectively.(1)$${DL}_{t}\%=\left(C_{0}-C_{t}\right)V/W_{C}*100\%$$(2)$${CE}_{t}\%=\left(C_{0}-C_{t}\right)/C_{0}*100\%$$(3)$$F_{t}\%=DL/{DL}_{\infty }*\;100\%$$(4)$$Y\%=W_{c}/(W_{r}+W_{d})*100\%$$
where *C*_0_ is the initial concentration of TRA in the solution, *C_t_* refers to the concentration of TRA at time *t*, *V* represents the volume of solution, *DL*_∞_ refers to the drug loading at equilibrium, and *W_c_*, *W_r_*, and *W_d_* are the weight of drug–resin complexes, resin, and drug, respectively.

### 2.3. Characterizations of Drug–Resin Complexes

#### 2.3.1. SEM-EDS

To visualize the surface morphologies of TRA, Amberlite IRP88, their physical mixtures (PMs), and TRCs_2:1_, a ZEISS Gemini 300 scanning electron microscope (Carl Zeiss AG, Oberkochen, Germany) equipped with an X-MaxN EDS detector (XPLORE30, Oxford Instruments, Oxford, UK) was employed. All samples were gold sputter-coated before observation. The applied accelerated voltage and resolution were set at 15 kV and 1 nm, respectively. The EDS images of the samples were recorded, with Cl and K detected and displayed in purple and light blue, respectively.

#### 2.3.2. PXRD

PXRD patterns were obtained using a D/MAX-2600 X-ray diffractometer (Rigaku Corporation, Tokyo, Japan) at a tube voltage of 40 kV with Cu Kα radiation. The current intensity was set at 40 mA. Diffraction patterns for TRA, Amberlite IRP88, their physical mixtures, and TRCs_2:1_ were collected over the range of 5–40° at 5°/min with a step size of 0.05°.

#### 2.3.3. Thermal Analyses

Thermal analyses were conducted using a DSC 250 instrument (TA Instruments, Delaware, Newcastle, USA). Approximately 5 mg of each sample was sealed in closed alumina crucibles, with an empty pan serving as the reference. DSC thermograms were recorded under a nitrogen atmosphere from 30 °C to 250 °C at a heating rate of 10 °C/min.

#### 2.3.4. FT-IR

FT-IR spectra were recorded using a Nicolet iS20 FTIR apparatus (Thermo Fisher Scientific Co., Ltd., Waltham, MA, USA) at a resolution of 2 cm^−1^. The spectral width was 500–4000 cm^−1^.

### 2.4. Molecular Docking

Molecular docking was conducted to explore the interaction between TRA and Amberlite IRP88 within TRCs. The initial Amberlite IRP88 and TRA structures were built and imported into Materials Studio. Then, the Smart Minimizer algorithm was selected to optimize the structures. The entire surface of the optimized Amberlite IRP88 was defined as the docking site. Next, docking simulations between TRA and Amberlite IRP88 were carried out using AutoDock 4.2.6. A docking box with an 80 Å × 80 Å × 80 Å grid was employed with a grid spacing of 0.375 Å. The number of conformational searches was set to 20, and other parameters were kept in default settings. Finally, the lowest energy conformation was optimized.

### 2.5. Dissolution Testing

Simulated saliva dissolution was performed in simulated saliva fluid (SSF) using a magnetic stirring apparatus (Shenzhen Labtemp Instrument Technology Ltd., Shenzhen, China). The SSF was prepared according to Huang’s study [36]. The SSF was prepared by dissolving potassium dihydrogen phosphate (0.534 g) in 120 mL of water and regulating the pH (6.8) using a hydrochloric acid and sodium hydroxide solution (1 mol/mL). The SSF was maintained at 37 ± 0.5 °C and stirred at 100 rpm. TRA, PM, and TRCs2:1 (equivalent to 50 mg of TRA) powders were directly poured into SSF (20 mL). After 30 s, 1.0 mL solution was withdrawn and centrifuged at 13,000 rpm for 3 min. The supernatant was determined at 246 nm using the UV spectrophotometer.

The dissolution tests of TCRs2:1 were also conducted in other mediums (purified water, 0.15, 0.3, and 0.6 mol/L NaCl solutions, pH 1.0 hydrochloric acid solutions, pH 4.5 acetate buffers, pH 6.8 and pH 7.4 phosphate buffers) using a DS1206 dissolution tester (Shenzhen Wahyong Analytical Instruments Co., Ltd., Shenzhen, China). Tests were carried out in 900 mL of medium at 37 ± 0.5 °C, and the paddle was set at 100 rpm. TRCs2:1 (equivalent to 50 mg of TRA) powders were directly poured into each vessel, and approximately 10 mL solution was withdrawn at 5, 10, 15, 20, 30, 45, 60, 90, and 120 min, followed by immediate replacement with an equal volume of fresh medium.

The dissolution samples were filtered through a 0.45 μm hydrophilic membrane filter (Millipore^®^ Corporation, Bedford, OH, USA). The filtrate was then appropriately diluted with dissolution medium to ensure the absorbance values fell within the range of 0.2–0.8 AU. The dissolution medium was used as a reference. The amount of TRA was quantified at 246 nm using a T-6 UV spectrophotometer (Nanjing Philes Instrument Co., Ltd., Nanjing, China) equipped with a photomultiplier tube detector and deuterium lamps.

### 2.6. Taste Evaluation

A total of 10 healthy adult volunteers (5 males and 5 females) participated in a human taste panel [37]. One hour before the tests, participants were not allowed to eat, drink (except water), smoke, or use oral hygiene products. Test samples (equivalent to 50 mg of TRA) were coded to ensure blinding. Participants followed a standardized procedure: (1) rinsing their mouth with water 3 times and waiting for 5 min, (2) placing the sample on their tongue for 30 s without chewing or swallowing, followed by spitting out, and (3) rating bitterness immediately on a numerical scale (0 = no bitterness, 1 = threshold bitterness, 2 = slight bitterness, 3 = moderate bitterness, 4 = bitterness, and 5 = extreme bitterness). The scores given by all participants were calculated as average taste scores. A lower score reflected a weaker bitterness.

### 2.7. Statistical Analysis

All experimental data were obtained at least in triplicate and expressed as the mean ± SD. Statistical differences were determined using Student’s two-tailed *t*-test. A *p*-value of less than 0.05 was considered statistically significant.

## 3. Results and Discussion

### 3.1. Preparation of TRCs

To elucidate the relationships between drug-to-resin ratios and the formation of complexes, three TRCs with varying TRA–resin ratios (1:2, 1:1, 2:1, *w*/*w*) were prepared. As shown in Figure 2A, the *DL* increased with an increase in the TRA-to-resin ratios. *DL* did not exceed 47% when the drug-to-resin ratio was 1:2 or 1:1, whereas a *DL* of 63% was achieved at the ratio of 2:1. This demonstrated that exceeding a threshold TRA concentration enhances its binding affinity and capacity on the Amberlite IRP88 resin, likely saturating available ion-exchange sites at lower ratios [38]. In addition, the *CE* and *F* values of all samples rose to approximately 90% and 100% at 120 min, respectively, and no significant differences (*p* > 0.05) were observed among the three formulations (Figure 2B and C). This indicated that, despite varying DL, the intrinsic ion-exchange process itself reached near-completion within the timeframe for all ratios. Additionally, *Y* of TRCs followed the order of TRCs_1:2_ (42.40%) < TRCs_1:1_ (56.49%) < TRCs_2:1_ (65.65%). This result suggested that the drug ratio had a positive effect on *Y* (Figure 2D). This trend aligned with the increased DL, suggesting that higher drug input directly translates to greater recoverable complex mass, albeit influenced by processing factors. Based on the optimization objectives prioritizing both high drug loading (DL) and overall yield (Y), TRCs_2:1_ was selected for further study.

### 3.2. Physical Characterizations of TRCs

#### 3.2.1. SEM-EDS

The morphologies of all samples are shown in Figure 3. Unprocessed TRA manifested as aggregates of irregular small block particles with rough surfaces, while Amberlite IRP88 possessed larger sizes and smoother surfaces than TRA. The PM simply displayed a physical admixture of the distinct morphologies associated with TRA and the resin. However, TRCs_2:1_ revealed a markedly altered morphology: no discrete, rough TRA particles were discernible within the field of view. This morphological homogenization strongly suggests the effective dispersion and integration of TRA within the resin matrix. To unequivocally confirm complex formation and ion exchange, EDS analyses were conducted [39]. As shown in Figure 3, TRA particles were identified by a strong chlorine (Cl) signal, depicted as purple in the overlay (Figure 3). Amberlite IRP88 was characterized by its dominant potassium (K) signal, rendered in light blue. Significantly, the PM image exhibited discrete purple (TRA) and light-blue (resin) particles with sharp boundaries. However, TRCs showed that each particle appeared both purple and light blue in the merged image. This intimate intermixing at the elemental level provides direct visual evidence that ion exchange has occurred: potassium counterions on the resin functional groups (-COO^−^K^+^) have been replaced by ionized TRA cations, forming complexes [38].

#### 3.2.2. Thermal Analyses

To elucidate the thermal properties of TRCs, DSC was conducted. As shown in Figure 4A, TRA exhibited a sharp endothermic melting peak at around 235 °C [40], characteristic of its crystal lattice breakdown. However, Amberlite IRP88 did not exhibit any endothermic peaks. In the DSC pattern of PM, the endothermic peak of TRA was still present, confirming the persistence of crystalline TRA domains within the mixture. Moreover, TRCs_2:1_ only exhibited a broad peak at 87 °C due to the adsorbed water, while the characteristic melting peak of crystalline TRA was entirely absent. This confirmed the transformation of the TRA crystalline structure to an amorphous form after the ion-exchange process.

#### 3.2.3. PXRD

The solid-state transformation of TRA within the complex was further investigated by PXRD. As shown in Figure 4B, the PXRD patterns of TRA and PM displayed characteristic diffraction peaks at 5.2°, 10.4°, 17.8°, 23.6°, and 24.4°. In contrast, Amberlite IRP88 did not show any peaks owing to its amorphous form. Additionally, TRCs_2:1_ showed complete absence of the characteristic TRA crystalline diffraction peaks, indicating a complete conversion of TRA from crystalline to the amorphous form. The results of PXRD demonstrated the successful formation of amorphous drug–resin complexes, which corroborated the DSC findings.

#### 3.2.4. FT-IR

The FT-IR spectra of TRA, Amberlite IRP88, PM, and TRAs_2:1_ are shown in Figure 4C. TRA showed strong absorption bands at 1702 cm^−1^ and 1098 cm^−1^, corresponding to C=O and C-Cl, respectively. The characteristic absorption peak at 2447 cm^−1^ was also observed due to the =N- stretching vibration in the tertiary amine group. Besides, TRA also displayed peaks at 1228 cm^−1^, 1257 cm^−1^, and 1273 cm^−1^, which represented the stretching vibration of C-N from the aromatic ring [41]. Amberlite IRP88 possessed an absorption peak at 1540 cm^−1^ (C=O) and 3223 cm^−1^ (-OH). The PM spectrum appeared as a superposition of individual TRA and resin spectra. Importantly, in the case of TRCs_2:1_, the absorption peak of the =N- group disappeared and showed blue-shifted peaks of C-N. This may be related to the restriction of vibrations due to the increased steric hindrance and the formation of hydrogen bonds during complexation [15,42]. Additionally, Han et al. considered that the variation peaks of the complex are closely tied to the formation of the salt bridge [20].

### 3.3. Molecular Docking

To elucidate the molecular basis of TRA–resin interactions, molecular docking was performed, and the corresponding image is shown in Figure 5. The yellow dashed lines indicate the formation of a charge-stabilized salt bridge between the protonated tertiary amine of TRA’s piperazine ring and the deprotonated carboxyl group of Amberlite IRP88. It revealed the reaction sites of the host–guest molecules in the ion-exchange process and explained the blue-shifted peaks of C-N observed in the FT-IR spectrum. Critically, salt bridges enhance complex stability through electrostatic anchoring. Previous studies establish that binding energies ≤ −8 KJ/mol signify stable complexes [43]. In our study, the binding energy of −12.32 KJ/mol indicated exceptional stability (exceeding the empirical threshold by >50%). This thermodynamic stability will directly translate to functional efficacy: the high energy barrier effectively inhibits complex dissociation in saliva, preventing TRA release and subsequent bitterness perception. Thus, based on the combined molecular and energetic evidence, it can be inferred that resin is a robust taste-masking carrier.

### 3.4. Dissolution Testing

To evaluate the potential for bitterness perception, dissolution profiles were quantified in SSF over 30 s. As shown in Figure 6A, pure TRA exhibited rapid dissolution (8.72% within 30 s), consistent with its high SSF solubility (46.99 mg/mL). In addition, PM showed a similar dissolution profile to TRA. On the contrary, only 1.52% of TRA was dissolved from TRCs_2:1_ within 30 s. This reduction in early-phase dissolution directly correlates with taste-masking efficacy. The observed inhibition aligns with prior reports where Amberlite IRP88 formed stable salt bridges with sildenafil citrate [23] and phencynonate HCl [20], significantly limiting drug liberation in salivary environments. Based on these results, it can be inferred that TRCs_2:1_ would not generate intense bitterness in the oral cavity.

Dissolution is a reversed process for the ion-exchange process [44,45], and drug release occurs through competitive displacement by counterions in the medium [44,46]. This mechanistic dependency implies that medium composition critically governs TRCs’ dissolution kinetics. First, the types of counterions are important, and the dissolution test was conducted in NaCl (0.15 mol/L), KCl (0.15 mol/L), and HCl (0.15 mol/L) solutions. As shown in Figure 6B, TRCs_2:1_ dissolved slowly in NaCl and KCl solutions but quickly in HCl solutions. This was because hydrogen ions had a stronger affinity to Amberlite IRP88 than other ions [23,39]. Secondly, ion strengths are also critical factors for the dissolution of complexes. As shown in Figure 6C, TRCs_2:1_ dissolved more slowly in water than that in NaCl solutions with different ionic strengths, because water contained fewer counterions [43,47]. However, when the concentration of NaCl increased from 0.15 mol/L to 0.60 mol/L, the dissolution rates of TRCs_2:1_ showed an increasing trend. These results indicated that the enhancement of ion strengths in the medium can promote the dissolution of TRCs_2:1_. Thirdly, the dissolution of TRCs_2:1_ was also dependent on the pH values of the medium. As shown in Figure 6D, the dissolution amount at 5 min followed the order of pH 1.0 (102.05%) > pH 4.5 (34.66%) > pH 6.8 (19.75%) > pH 7.4 (8.29%). This was because the pH 1.0 medium had more hydrogen ions and a stronger ability to replace ionized TRA than other ions. Notably, almost 100% of TRCs_2:1_ was dissolved in pH 1.0, 4.5, and 6.8 buffers over 2 h. This was because TRA was amorphously dispersed in Amberlite IRP88, which enabled rapid dissolution. Thus, it can be inferred that TRCs_2:1_ can be well absorbed in the gastrointestinal tract.

### 3.5. Taste Evaluation

To assess the taste-masking ability of amorphization, the bitterness of all samples was evaluated by ten healthy adult volunteers, and the results are presented in Table 1. It was found that pure TRA generated intense bitterness (average bitter score = 5.0) immediately after it was placed on volunteers’ tongues. On the contrary, all volunteers suggested that Amberlite IRP88 did not have any bitter taste. Moreover, the average bitter score of the PM reached 4.6 and did not have a significant difference with pure TRA (*p* > 0.05). This suggested that simply mixing the drug with the resin cannot mask the bitterness of TRA. Importantly, only four out of ten volunteers indicated that TRCs_2:1_ possessed threshold bitterness, and its average bitter score was as low as 0.5. These results suggested that the bitterness of TRA was successfully masked by Amberlite IRP88. This success was mechanistically attributed to two aspects. On the one hand, TRA was amorphously encapsulated in Amberlite IRP88 resin, which could prevent penetration of saliva and thus avoid significant drug dissolution. On the other hand, the stabilized salt bridge between the resin and the drug can effectively inhibit the dissociation of the complex in saliva.

Although significant taste improvement of TRA–resin complexes was demonstrated in healthy adults, this formulation warrants cautious extrapolation to pediatric populations due to inherent differences in taste perception. Therefore, future studies will focus on evaluating the palatability of formulations in pediatric subjects. In addition, a “threshold bitterness” was still perceived by some participants. Thus, it is important to employ complementary methods to further suppress the bitterness of TRA. For example, γ-aminobutyric acid [48] and phosphatidic acid [49] are bitter taste receptor antagonists, which can prevent the generation of bitterness. Additionally, addition of sweeteners (e.g., mannitol and sucrose) to formulations can impart sweetness and thus effectively neutralize residual bitterness [50]. Therefore, a strategy combining the drug–resin complexes with bitter taste receptor antagonists and sweeteners has potential for achieving optimal palatability in the future.

## 4. Conclusions

In this study, we developed a palatable amorphous TRA formulation using Amberlite IRP88 as the carrier. Molecular docking elucidated a salt bridge between the nitrogen atom on the piperazine ring in TRA and the carboxyl group in Amberlite IRP88 was formed during the ion-exchange process. In addition, the results of simulated saliva dissolution indicated that TRCs only dissolved 1.52% of TRA in SSF, and taste evaluation suggested that the bitterness of TRA was successfully masked after complexing with Amberlite IRP88. In summary, amorphization based on ion exchange is an effective strategy to mask the bitterness of drugs.

## Figures and Tables

**Figure 1 pharmaceutics-17-00972-f001:**
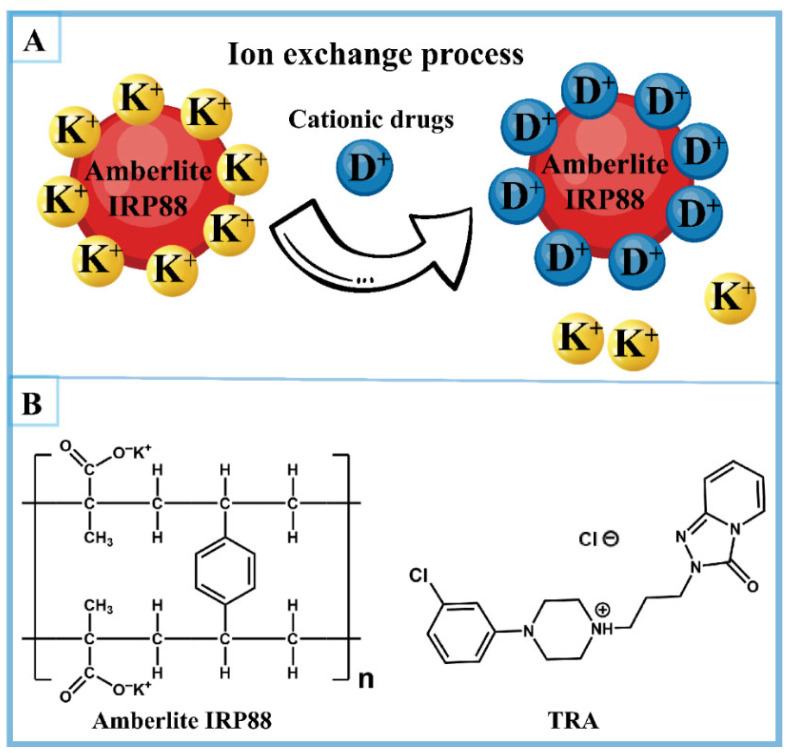
(**A**) Ion-exchange processes between Amberlite IRP88 and cationic drugs and (**B**) the structures of Amberlite IRP88 and TRA.

**Figure 2 pharmaceutics-17-00972-f002:**
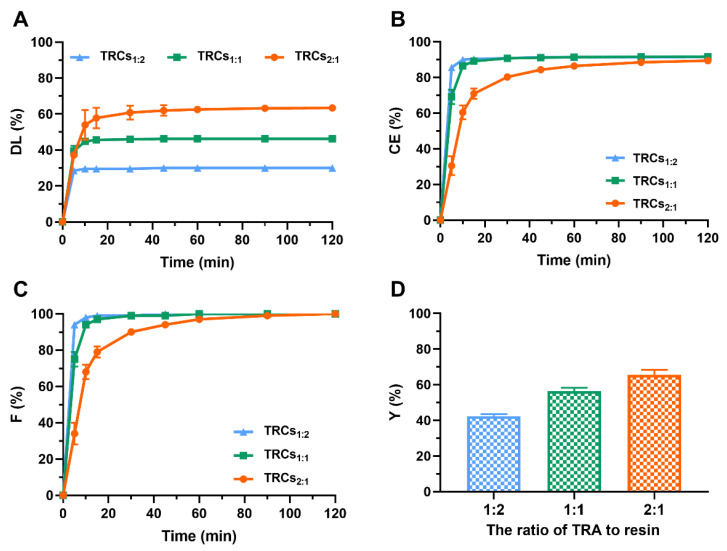
Influences of the drug-to-resin ratios on (**A**) *DL*, (**B**) *CE*, (**C**) *F*, and (**D**) *Y* (data are mean ± SD, n = 3).

**Figure 3 pharmaceutics-17-00972-f003:**
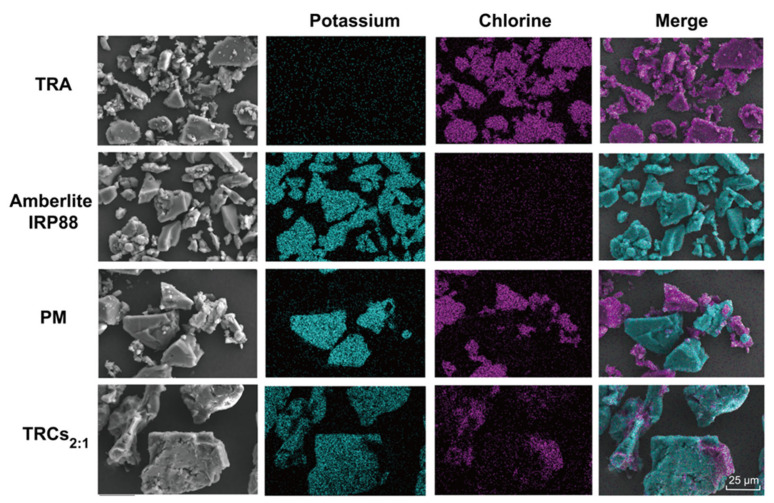
SEM-EDS images of potassium and chlorine elements captured from TRA, Amberlite IRP88, PM, and TRCs_2:1_.

**Figure 4 pharmaceutics-17-00972-f004:**
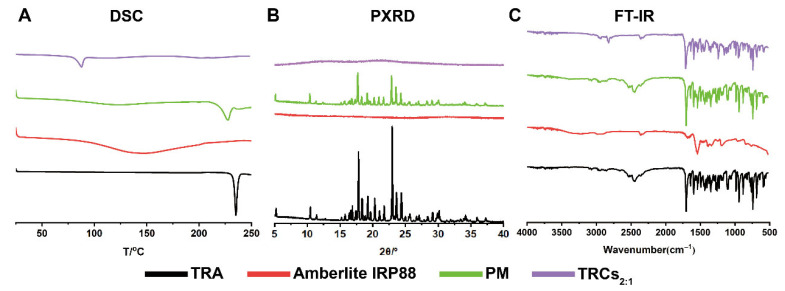
Physical characterization of TRA, Amberlite IRP88, PM, and TRCs_2:1_: (**A**) DSC, (**B**) PXRD, and (**C**) FT-IR.

**Figure 5 pharmaceutics-17-00972-f005:**
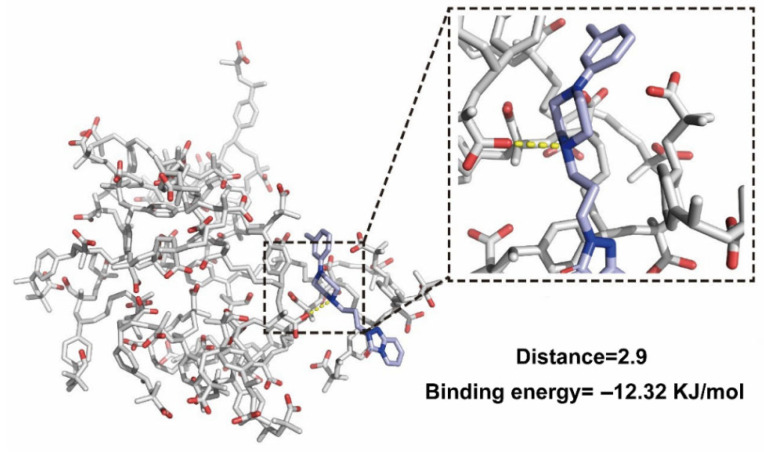
The molecular docking image of TRCs. Oxygen atoms are in red, and nitrogen atoms are in dark blue.

**Figure 6 pharmaceutics-17-00972-f006:**
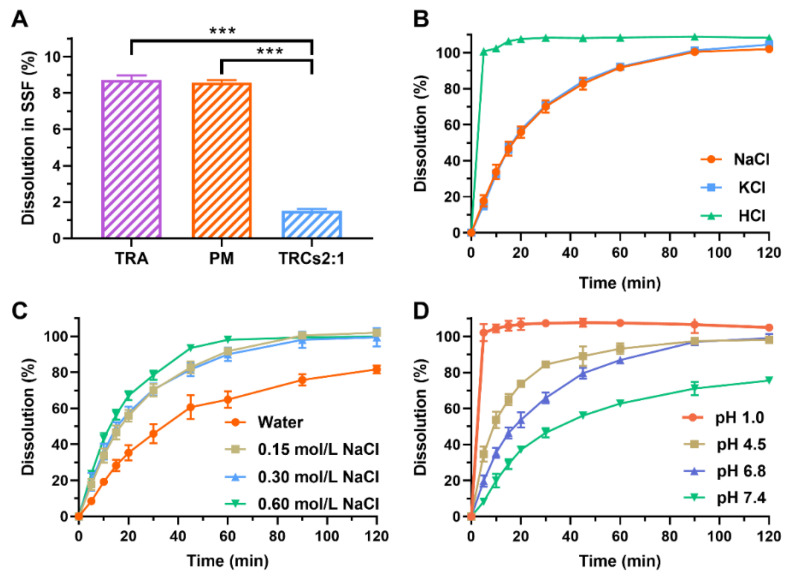
Dissolution testing in SSF (**A**). Influences of types of counterions (**B**), ion strengths (**C**), and pH values of media (**D**) on dissolution profiles of TRCs_2:1_ (n = 3, mean ± SD, *** represents *p* < 0.001).

**Table 1 pharmaceutics-17-00972-t001:** Bitter score of TRA, Amberlite IRP88, PM, and TRCs_2:1_.

Participants	TRA	Amberlite IRP88	PM	TRCs_2:1_
1	5	0	4	0
2	5	0	5	1
3	5	0	5	0
4	5	0	4	1
5	5	0	5	0
6	5	0	5	2
7	5	0	4	0
8	5	0	5	0
9	5	0	4	0
10	5	0	5	1
Sum	30	0	46	4
Average	5.0	0.0 ***	4.6	0.5 ***
SD	0.0	0.0	0.5	0.7
RSD (%)	0.0	\	0.1	1.4

*** represents *p* < 0.001 compared with the pure TRA. 0 = no bitterness, 1 = threshold bitterness, 2 = slight bitterness, 3 = moderate bitterness, 4 = bitterness, and 5 = extreme bitterness.

## Data Availability

The original contributions presented in this study are included in the article/Appendix A. Further inquiries can be directed to the corresponding author(s).

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
