# Peer review of "Development of Palatable Amorphous Trazodone Hydrochloride Formulations via Ion Exchange"

_pharmaceutics, 2025, doi:10.3390/pharmaceutics17080972_

Round 1

Reviewer 1 Report

Comments and Suggestions for Authors

The current research article focusing on the development of palatable trazodone by forming a complex with resin lacks novelty. The authors are suggested to address the below comments:

  1. Can authors please rewrite the conclusion present in the abstract to make it clear if taste masking was achieved due to presence of drug in amorphous state or due to complexation with the resin?
  2. Can authors please justify the reason for selecting trazodone hcl as a bitter active substance?
  3. Is there need for converting drug from crystalline to amorphous state. Considering salt form of the drug it should have good solubility.
  4. For tablet which is meant to disintegrate in stomach, why was unpleasant taste of drug much important? A simple film coating of tablets should be sufficient enough to prevent contact of drug with sensory receptors present in the stomach.
  5. SEM analysis will provide an understanding of surface morphology of the material but will not provide any information about the formation of drug-resin complex. Please correct the methodology as needed
  6. Please provide a detailed UV-methodology rather than just providing the wavelength
  7. How were the samples added to dissolution vessel?
  8. What was the pH of SSF
  9. Please provide the information for ethical approval for conducting in-vivo studies
  10. The results section needs to be improved with more scientific justification rather than superficial discussion.
  11. The entire article needs to be corrected for grammatical mistakes and missing articles.

Reviewer 2 Report

Comments and Suggestions for Authors

The authors aimed to develop a palatable amorphous formulation of trazodone hydrochloride (TRA) using Amberlite IRP88 resin through a static ion exchange method. 

To achieve higher drug loading efficiencies, it is preferable for future study or a more thorough discussion to look into different resin types or examine further optimization parameters (such as changing temperature, stirring speed, or initial TRA concentration beyond the measured range).

The introduction strongly emphasizes the importance of taste masking for pediatric patients. However, the in vivo taste evaluation was conducted solely on adults. Taste perception and drug acceptance can differ significantly in children. Also, the human taste panel consisted of only 10 healthy adult volunteers. While the results are statistically significant, the small sample size may limit the generalizability of the findings due to inherent variability in human taste perception.

While TRCs 2:1 showed significant improvement, four out of ten volunteers still perceived "threshold bitterness." The authors could discuss strategies to further minimize this residual bitterness, especially for highly sensitive populations.

In Figure 5, the binding energy is stated as "12.32 KJ/mol". However, in the discussion, it is mentioned that complexes are stable when binding is "below -8~KJ/mol". This creates an inconsistency in the sign convention for binding energy. Binding energy is typically negative for favorable (stable) interactions.

Figure 6C Legend: The legend for Figure 6C includes an entry for "pH 7.4 (8.29%)" which implies a corresponding curve or data set, but no such data is visibly represented in the graph.

Reviewer 3 Report

Comments and Suggestions for Authors

The authors presented development of palatable amorphous trazodone hydrochloride formulations using ion exchange process. The study is well designed and could be interesting for readers. However, I think the manuscript should be corrected prior to acceptance and here are my comments and suggestions:

  1. In the introduction section, the authors should give more details on trazodone (about his pharmacological application and also about the formulations that can be found on the market)
  2. Also in introduction, line 61: carboxylic group cannot be replaced by cationic drugs but K+ ion is replaced. Please correct this to avoid misunderstandings.
  3. Section 2.4 (molecular docking) - this section requires more experimental details. What is the minimization method used for the preparation of 3D structures of molecules? What part of the polymer is used for docking or the docking box covers the whole polymer?
  4. Section 2.6.1 - why is dissolution in simulated saliva placed within taste evaluation (section 2.6)? From my point of view, it is logical to place this dissolution experiment together with other dissolution experiments (in the section 2.5)
  5. Section 3.2.1 - please explain the principles of EDS and why different ions are differently colored?
  6. Section 3.5.1 - similarly to one of my previous comments. It seems more logical not to place dissolution in simulated saliva together with the taste evaluation

Round 2

Reviewer 1 Report

Comments and Suggestions for Authors

All the comments are well addressed with proper justification and supporting literature. The revised version of the article can be accepted for publication

Reviewer 3 Report

Comments and Suggestions for Authors

The authors corrected manuscript in accordance to my comments and, in this form, it can be accepted for publication